# Reasons for Discordance between ^68^Ga-PSMA-PET and Magnetic Resonance Imaging in Men with Metastatic Prostate Cancer

**DOI:** 10.3390/cancers16112056

**Published:** 2024-05-29

**Authors:** Jade Wang, Elisabeth O’Dwyer, Juana Martinez Zuloaga, Kritika Subramanian, Jim C. Hu, Yuliya S. Jhanwar, Himanshu Nagar, Arindam RoyChoudhury, John Babich, Sandra Huicochea Castellanos, Joseph R. Osborne, Daniel J. A. Margolis

**Affiliations:** 1Department of Internal Medicine, New York-Presbyterian Hospital, New York, NY 10065, USA; 2Department of Radiology, Weill Cornell Medical College, New York, NY 10065, USA; 3Department of Urology, Weill Cornell Medical College, New York, NY 10065, USA; 4Department of Radiation Oncology, Weill Cornell Medical College, New York, NY 10065, USA; hnagar@med.cornell.edu; 5Department of Population Health Sciences, Weill Cornell Medical College, New York, NY 10065, USA; 6Ratio Therapeutics, Inc., Boston, MA 02210, USA

**Keywords:** prostate, cancer, MRI, PET, PSMA, whole-body

## Abstract

**Simple Summary:**

MRI uses magnetic pulses to create images of the body. PET scans use radioactive chemicals to determine what kinds of processes are going on in the body. PSMA is a chemical that is abundant on prostate cancer cells, so a PET scan using a radioactive chemical that attaches to PSMA shows where prostate cancer is, but this test has drawbacks: it is not widely available, it is expensive, and it exposes patients to radiation. However, a recent study found that PSMA PET was more accurate than MRI for finding areas where prostate cancer spreads. This study looks at the “false negative” cases—the specific cases where the MRI did not find prostate cancer when PSMA PET did—and how reading MRI can be improved.

**Abstract:**

Background: PSMA PET has emerged as a “gold standard” imaging modality for assessing prostate cancer metastases. However, it is not universally available, and this limits its impact. In contrast, whole-body MRI is much more widely available but misses more lesions. This study aims to improve the interpretation of whole-body MRI by comparing false negative scans retrospectively to PSMA PET. Methods: This study was a retrospective sub-analysis of a prospectively collected database of patients who participated in a clinical trial of PSMA PET/MRI comparing PSMA PET and whole-body MRI from 2018–2021. Subjects whose separately read PSMA PET and MRI diagnostic reports showed discrepancies (“false negative” MRI cases) were selected for sub-analysis. The cases were reviewed by the same attending radiologist who originally read the scans. The radiologist noted specific features on MRI indicating metastatic disease that were initially missed. Results: Of 263 cases, 38 (14%) met the inclusion criteria and were reviewed. Six classes of mpMRI false negatives were identified: anatomically normal (18, 47%), atypical MRI appearance (6, 16%), mischaracterization (1, 3%), undercall (6, 16%), obscured (4, 11%), and no abnormality on MRI (3, 8%). Considering that the atypical and undercalled cases could have been adjusted in retrospect, and that 4 additional cases had positive lesions to the same extent and 11 further cases had disease confined to the pelvis, only 11 (4%) of the original 263 would have had disease outside of a conventional radiation treatment plan. Conclusion: Notably, almost 50% of the cases, including most lymph node metastases, were anatomically normal using standard criteria. This suggests that current anatomic criteria for evaluating prostate cancer lymph node metastases are not ideal, and there is a need for improved criteria. In addition, 32% of cases involved some element of human interpretive error, and, therefore, improving reader training may lead to more accurate results.

## 1. Introduction

Prostate cancer is the most common cancer in men and the second leading cause of male cancer mortality in the United States [1]. The lifetime risk of developing prostate cancer for a man living in the United States is estimated at 1 in 8. While prostate cancer often has a relatively indolent course, metastatic disease is not uncommon and carries a high mortality burden. The 5-year survival rate for patients with only local or regionally advanced prostate cancer is close to 100%, while the 5-year survival rate for patients with metastatic disease is only 30% [2].

Thus, it is clinically important to assess the presence and extent of metastasis in patients with prostate cancer. This has a major impact on treatment planning: patients with early-stage disease are eligible for surgical excision and localized radiotherapy, while patients with metastatic disease are generally limited to chemotherapy or androgen-deprivation therapy [3,4].

The assessment of prostate cancer metastasis has traditionally been performed by radionuclide bone scan and CT. However, these imaging methods cannot always accurately detect non-localized disease, resulting in disease recurrence even in patients initially treated with curative intent [5,6]. Meanwhile, magnetic resonance imaging (MRI) has been gaining increased recognition as a useful method of evaluating lymph nodes and distant metastases in patients with various cancers, including prostate cancer [7,8]. Yet, MRI, despite its advances, also still misses clinically relevant metastatic disease.

In recent years, a new imaging method has emerged for primary staging and assessment of metastatic prostate cancer. ^68^Ga-PSMA-HBED-CC positron emission tomography (PET) imaging uses gallium-radiolabeled prostate-specific membrane antigen (PSMA), a peptide largely specific to prostatic tissue, including prostate cancer [9]. Although the spatial resolution of PET is lower than MRI, with lesions smaller than 6 mm not reliably detected, this is offset by its superior contrast-to-background. The major limitation of PSMA PET is that it requires the administration of novel radiotracers such as ^68^Ga-PSMA-11. As a result, it is not widely available outside of large, resource-rich medical centers equipped with adequate infrastructure, and this limits its utility. Especially given that there is an order of magnitude more MRI units compared with PET units in the United States, with many states having only two or fewer centers providing this service and with many countries in the “global south” lacking PET altogether, one cannot count on PSMA PET availability despite the FDA having approved more than one formulation [10,11,12]. In contrast, while MRI is much more widely available, it misses a higher percentage of lesions compared to PSMA PET [13].

There is therefore a valuable opportunity to use PSMA PET as a standard of reference to improve the interpretation of MRI. This project aims to investigate whether the ability of MRI scans to detect metastatic prostate cancer can be enhanced using retrospective information provided by concomitant PSMA PET scans. To our knowledge, such an approach has not yet been taken. Improving the interpretation of MRI will allow more timely and effective diagnosis of metastatic prostate cancer in settings that lack PSMA PET imaging, and it will greatly benefit patients by allowing them to receive earlier treatment.

## 2. Materials and Methods

This study was a sub-analysis of an IRB-approved, HIPAA-compliant prospectively collected database of 263 patients who provided written informed consent for and participated in a clinical trial of ^68^Ga-PSMA-HBED-CC PET/MRI comparing separately read PSMA PET and whole-body MRI and multiparametric MRI (mpMRI) of the prostate/prostatic bed from 2018 and 2021 at a single institution. Patients underwent concomitant PSMA PET/MRI for one of two general indications: (1) treatment planning or (2) biochemical recurrence based on serum prostate specific antigen (PSA) levels (defined either as PSA ≥ 0.2 ng/mL in radical prostatectomy patients or as 2 ng/mL greater than the nadir PSA in post-radiation therapy patients). Demographics are listed in Table 1. A formal description of the PSMA PET scanning technique has been previously reported, and 30 of the subjects analyzed herein overlap with this prior study, which focused only on subjects with biochemical recurrence [14].

In each case, the PSMA PET was read without first reviewing the MRI and vice versa, and the results were recorded. Subjects whose PSMA PET and MRI diagnostic reports showed discrepancies were selected for sub-analysis. Inclusion criteria included all patients whose PSMA PET was positive for metastases in the lymph node(s), bone, and/or other tissues while the corresponding MRI was negative. 

Cases were excluded if (1) data were missing for one or both scans; (2) data could not be interpreted for one or both scans; (3) the MRI was found upon review to have correctly identified the lesion in question; or (4) the PSMA PET was a false positive based on biopsy results (with no subjects in this last category). See Figure 1.

Relevant cases were reviewed by the same attending radiologist who originally read the scans. The attending radiologist reviewed both the PSMA PET and MRI images and noted any common or stereotypical features on MRI (e.g., lesion size, location, or morphology) indicating metastatic disease that was initially missed. Six classes of false negative MRI were considered (Table 2).

## 3. Results

Of the 263 subjects in the database, we identified 38 (14%) “MRI false negative” subjects that both met inclusion criteria and were not excluded by exclusion criteria. The average age at time of scan was 69 years (range 46–88). The maximum PSA at time of scan was recorded for 33 of 38 patients and ranged from 0.21–143 ng/mL. 

The indications for imaging were the following (number of patients, %): conventional imaging equivocal or suggestive of prostate cancer metastatic disease (2, 5.2%); elevated PSA with no conventional imaging suggestive of metastatic or recurrent disease (20, 52.6%); planned for surgical extirpation (high-risk primary disease) (3, 7.9%); planned serial follow-up (i.e., focal therapy) with or without radiation therapy (4, 10.5%); planned for targeted biopsy of primary lesion (1, 2.6%); and other (8, 21.0%).

After the PSMA PET/MRI and mpMRI images were reviewed for each case, the 38 MRI false negative cases were found to fall into six broad classes. The common features associated with each class are outlined below in Table 3 and Figure 2.

Upon further review, of those cases that were “false negative” on MRI, other sites of disease were found such that the degree of extent (local, regional, or distant) would not change in 4 cases (11%), such that management would not be altered. Additionally, 12 cases (32%) were either atypical or “undercalled”, such that a change in reporting could have resulted in these cases being correctly classified. Regardless, the remaining 22 cases (58%) may have had disease not otherwise reported. However, of these cases, half (11, or 29% of 38) would have disease that would not have been included in a traditional pelvic lymph node and prostate bed radiation field. Compared to the entire dataset of 263 subjects, this results in 4% with undiagnosed extrapelvic disease.

### 3.1. Example 1—Anatomically Normal Lymph Node Metastasis

As seen in Figure 3, the largest class, comprising half of the cases, was anatomically normal. In other words, the size and morphology of the metastatic focus did not suggest an abnormality on MRI. This description applied to many false negative cases involving lymph node metastases. In this case, the initial mpMRI was negative for the lymph node metastases by size criteria, while the PSMA PET showed a PSMA-avid, subcentimeter, left-sided lesion (Figure 3C). Detailed review of the mpMRI confirmed that the lymph node in question measured well under 1.0 cm in short axis, with a similar “cold” contralateral lymph node with essentially the same characteristics (Figure 3A,B). There were no other features in the mpMRI that were suspicious for metastatic disease.

### 3.2. Example 2—Mischaracterization of Lymph Node Metastasis

Among the cases included in this analysis, there was only one example of a characterization issue, defined as an instance where a lymph node was suspicious on MRI on the basis of size but was not prospectively included in the report (Figure 4). This case involved a lymph node metastasis. The PSMA PET showed a PSMA-avid node in the left supraclavicular region (Figure 4C). The lymph node measured > 1.0 cm in short axis (Figure 4A). However, it also exhibited a benign morphology with an oval shape and a fatty hilum.

In terms of “atypical” and “undercalled” findings, with six of each, there were four “atypical” bone lesions: two rib lesions, one with signal loss but no T2 lengthening or restricted diffusion, and another with high signal on fat-suppressed T1/T2 but low signal on native T1/T2 and possible diffusion-restriction vs. T2 shine-through; one case of sternal subtle T1/T2 lengthening and possibly diffusion restriction also attributed to T2 shine-through; and one case of ischial T2-lengthening with an equivocal defect on T1 but no diffusion restriction. The other two “atypical” cases included asymmetric enhancement without restricted diffusion or a discrete mass in the seminal vesicles after radiation therapy, and thoracic lymph nodes that were slightly distorted from susceptibility artifact. The “undercalled” cases included five cases of lymph nodes that did not meet the size criteria but were either considered equivocal on the basis of round shape or short axis > 5 mm in the pelvic “landing zone” distribution, and one case where not all of the osseous lesions were identified. Obscured lesions included three cases of lymph nodes obscured by susceptibility artifact, either from lungs or metal, and one case where extensive degenerative bony change obscured an underlying focus that was PSMA-avid.

## 4. Discussion

This study aimed to classify major causes of MRI “misses” of metastatic prostate cancer when compared to PSMA PET, which was used as the standard of reference. In our analysis, the primary reason (47% of cases) why MRI missed metastatic lesions that were radiotracer-avid on PSMA PET/MRI was simply due to the lesions’ normal appearance by existing anatomic size criteria. Although MRI has been shown to be comparable in sensitivity and specificity to some forms of traditional PET/CT, such as FDG-PET/CT and F18-choline PET/CT, and it is one of the most common methods of evaluating prostate cancer metastases, MRI interpretation still relies heavily on size as the main criteria for the evaluation of lymph nodes [15,16].

Our results suggest that the current standards for evaluating potentially metastatic prostate cancer lesions, particularly for metastatic lymph nodes, are not ideal. Current practice generally considers oval lymph nodes ≥ 1.0 cm in short axis or round nodes ≥ 0.8 cm in diameter as indicators of metastasis. However, these guidelines are optimized for increased specificity, not sensitivity [17]. As a result, MRI fairly frequently misses small metastases in “normal”-sized lymph nodes [13,18]. In contrast, PSMA PET, because it relies on PSMA expression and not solely on size criteria, shows superior performance in situations where metastatic lesions are more likely to be small. For example, PSMA PET has been demonstrated to have higher sensitivity than MRI in patients with low PSA values [19], as well as in patients with biochemical recurrence following radical prostatectomy [14]. Guberina et al. also found that PSMA PET/MRI was superior in detecting locally recurrent disease when compared to PSMA PET/CT [20].

Given the fact that MRI more easily misses smaller lesions using anatomic criteria alone, there is a need for multiparametric criteria to identify lesions that are functionally or biologically abnormal, instead of using purely size-based methods. One such alternative criteria has been proposed by Conlin et al. [21] Their group suggests using a four-compartment signal model of whole-body diffusion-weighted imaging, which appears to have promising potential to evaluate prostate cancer bone metastases

Additionally, 32% of the false negative cases analyzed were either characterization issues, atypical, or undercalled. In these cases, the lesions in question were initially seen on MRI but not given enough weight, often because they appeared in unexpected contexts. The one mischaracterized lesion, while significant by the size criteria, had a physiologic shape. Moreover, the supraclavicular region is an unusual location for solitary prostate cancer metastasis. These unusual features may have resulted in the initial interpretation of the lymph node as non-metastatic. Since there is some element of human interpretive error involved, improving reader training in terms of gauging suspicion may lead to more accurate results. The potential exists for clinical suspicion to inform the degree to which lesions are deemed suspicious on MRI.

The single case of “mischaracterization”, where the single-reader failed to identify a lymph node in an atypical location but where the lymph note was clearly suspicious based on the size criteria, falls within normal radiology performance given the 263 prospectively enrolled subjects. The veracity of this finding could be reinforced by having a dual-reader or repeat single-reader analysis for inter-rater versus intra-rater assessment, respectively, but this was not part of the experimental design.

Ultimately, it is crucial to improve MRI interpretation because PSMA PET, though undoubtedly more powerful and superior to traditional methods, is currently limited by availability to high-resource regions [10,11,12]. PSMA imaging requires the use of radiotracers, which must be produced through a relatively laborious manufacturing process and which have relatively short half-lives [22,23,24,25]. Improving the interpretation of mpMRI has significant potential to positively impact the diagnosis and treatment of metastatic prostate cancer for many patients in developing and sparsely populated regions.

Future avenues of investigation include a deeper review of pulse sequence-specific features (e.g., diffusion restriction) to determine whether subtle pulse-specific abnormalities that indicate metastatic disease are present on the MRI. If present, such features may reveal new abnormalities in “normal” cases to help with diagnosis. Additionally, such features could be a valuable resource to improve reader training for atypical and potentially undercalled cases.

This study was limited by a fairly small sample size; out of 263 PSMA PET/MRI scans, only 38 cases (14%) had a false negative MRI result compared to PSMA. In addition, the cases consisted of patients with diverse clinical circumstances—some planning for initial therapy, others who had already received surgical and/or radiation therapy. This somewhat heterogenous sample size may also complicate retrospective MRI interpretation and could introduce bias in terms of analysis from a skewed population. Additionally, some of these cases (e.g., planning for focal therapy) might not normally entail whole-body MRI were it not for their participation in the prospective clinical trial. Whole-body MRI remains a relatively uncommon procedure, and while it holds the potential to be more widespread than PET scanning, issues including costs, time requirements, and expertise limit its value. Lastly, there is a possibility that some cases were “PSMA false positive” as pathologic confirmation was only available for a minority of cases.

## 5. Conclusions

In conclusion, it is rare that MRI will miss metastatic disease caught by PSMA PET, and rarer still that, if one does not consider atypical and equivocal cases falsely negative, only 4% of this cohort would have disease missed by conventional treatment planning. Regardless, improving multiparametric characterization of small lymph nodes holds the greatest promise to compensate for this, the most common class of false negative MRI.

## Figures and Tables

**Figure 1 cancers-16-02056-f001:**
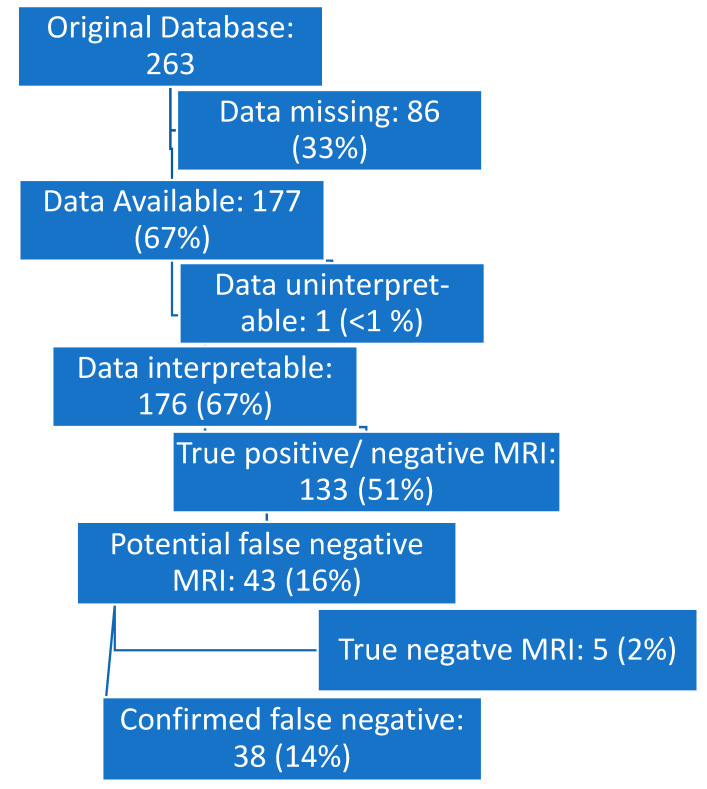
Flow diagram.

**Figure 2 cancers-16-02056-f002:**
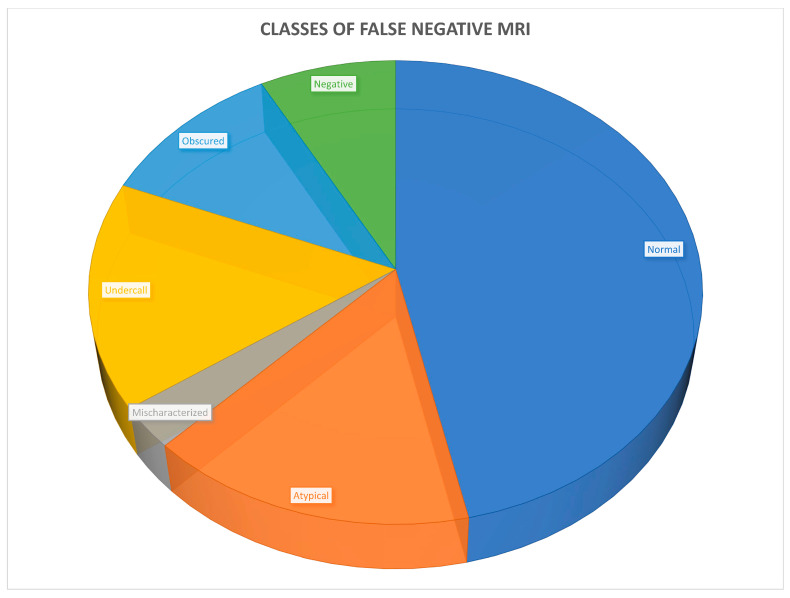
Six classes of MRI false negatives.

**Figure 3 cancers-16-02056-f003:**
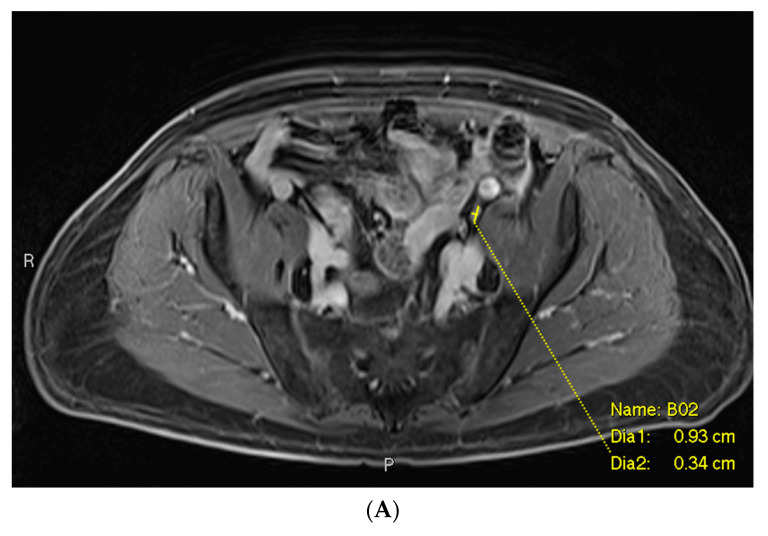
Post-contrast fat-saturated gradient T1-weighted image (**A**), diffusion-weighted imaging with b = 900 s/mm^2^ (DWI) (**B**), and fused PET/single-shot T2-weighted imaging (**C**) through the pelvis. In Figure 3A, the lesion is marked with measurement calipers. In Figure 3B,C, the lesion is denoted with a red arrow. Yellow arrows indicate the ureters, which show physiologic PSMA PET uptake.

**Figure 4 cancers-16-02056-f004:**
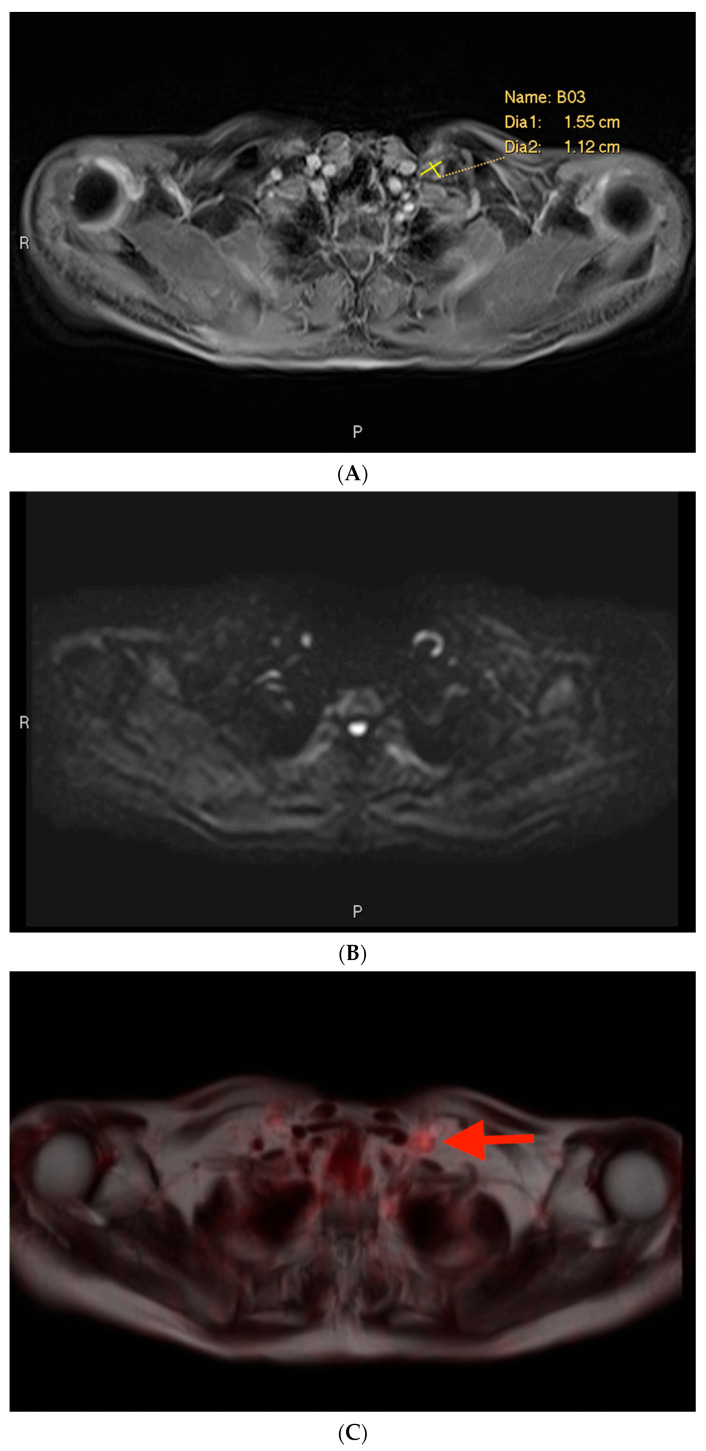
Post-contrast fat-saturated gradient T1 (**A**), DWI (**B**), and fused PET (**C**) images. In Figure 4A, the lesion is retrospectively measured (calipers). In Figure 4C, the lesion is denoted with a red arrow.

**Table 1 cancers-16-02056-t001:** Demographics.

Subjects	263 Total	38 Met Criteria
Indications	(analyzed subjects only)	
	BCR, conventional imaging equivocal/suggestive for metastasis	2 (5.2%)
	BCR, conventional imaging negative for metastasis	20 (52.6%)
	High-risk primary cancer for surgical planning	3 (7.9%)
	Radiation/focal therapy planning for primary cancer	4 (10.5%)
	Biopsy planning for primary lesion	1 (2.6%)
	Other indication (not specified)	8 (21.0%)
Age	Average 69 years	range 46–88
Serum PSA	range 0.21–143 ng/mL	
Ethnicity reported	62%	
Hispanic/Latino	3%	
Race reported	50%	
White	91%	
Black/African American	6%	
Asian/Pacific Islander	3%	

**Table 2 cancers-16-02056-t002:** False negative classes.

False Negative Class (Number of Cases, %)	Description
Anatomically normal	This type of false negative occurred exclusively with lymph node metastases. The affected lymph node showed PSMA PET uptake but was anatomically normal in size (<10 mm in short axis for oval/reniform nodes or <8 mm in diameter for spherical nodes) and physiologic in shape on mpMRI.
Atypical MRI appearance	mpMRI appearance was abnormal but not suspicious and was not prospectively described.
Mischaracterization	The lesion was suspicious on mpMRI but was not initially reported.
Undercall	mpMRI appearance was abnormal but was described as “equivocal” or “low suspicion”.
Obscured	Relevant area was difficult was to visualize on MRI because of technical or patient factors (e.g., hip replacement).
Negative	There was no focal structure on MRI to correlate with PSMA PET uptake.

**Table 3 cancers-16-02056-t003:** Breakdown of the six classes of MRI false negative.

False Negative Class (Number of Cases, %)
Anatomically normal (18, 47%)
Atypical MRI appearance (6, 16%)
Mischaracterization (1, 3%)
Undercall (6, 16%)
Obscured (4, 11%)
Negative (3, 8%)

## Data Availability

Data are not available for external review per institutional subject protection requirements.

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
