# Peer review of "Reasons for Discordance between 68Ga-PSMA-PET and Magnetic Resonance Imaging in Men with Metastatic Prostate Cancer"

_cancers, 2024, doi:10.3390/cancers16112056_

Round 1

Reviewer 1 Report

Comments and Suggestions for Authors

This manuscript describes an important analysis of the origin of false negative findings of current MP-MRI for metastatic disease (as compared  with 68Ga-PSMA PET). The authors describe strategies to decrease the false negative rates of MP-MRI which would be an important, clinically relevant, advance. Some minor revisions could improve this manuscript. 

 Modify the Title; Reasons for Discordance Between 68Ga-PSMA-PET and Magnetic Resonance Imaging in Men with Metastatic Prostate Cancer

Add a discussion of the spatial resolution of both 68Ga-PSMA-PET and MP-MRI to the introduction.

Author Response

Dear MDPI Cancers Editors and Staff and Reviewers,

On behalf of my co-authors, I would like to thank you for taking the time to review our submission and provide comments. We hope that you will find it significantly improved, in no small part because of your thoughtful comments.

This manuscript describes an important analysis of the origin of false negative findings of current MP-MRI for metastatic disease (as compared  with 68Ga-PSMA PET). The authors describe strategies to decrease the false negative rates of MP-MRI which would be an important, clinically relevant, advance. Some minor revisions could improve this manuscript.

Modify the Title; Reasons for Discordance Between 68Ga-PSMA-PET and Magnetic Resonance Imaging in Men with Metastatic Prostate Cancer

We appreciate this suggestion and have modified the title as suggested in the title page and body. However, we must defer to the Editors for the final decision on the revised title.

Add a discussion of the spatial resolution of both 68Ga-PSMA-PET and MP-MRI to the introduction.

Another good point. This has been added to the Introduction, 3rd paragraph: “Although the spatial resolution of PET is lower than MRI, with lesions smaller than 6 mm not reliably detected, this is offset by its superior contrast-to-background.”

Reviewer 2 Report

Comments and Suggestions for Authors

The retrospective study, “Reasons for Discordance Between 68Ga-PSMA-PET and Magnetic Resonance Imaging in Men with Prostate Cancer” attempts to identify the cause of false negative MRI findings compared to PSMA PET in men with prostate cancer undergoing PSMA PET/MRI.  While this analysis could have utility to the reader and the field in the setting of local disease, visceral disease (e.g liver metastasis) or osseus lesions, the authors were only able to identify nodal metastatic lesions as the site of false negative finding on MRI.  The improved sensitivity and accuracy of PSMA PET compared to traditional imaging for nodal disease is a well-known and established aspect of molecular imaging (including other molecular PET agents for prostate cancer such as 11C-choline and 18F-fluciclovine) in that nodal metastatic disease is able to be identified on PSMA PET prior to pathological enlargement.  This is in part the excitement and adoption of PSMA PET/CT throughout the world.  Thus, while this analysis supports already available information in the literature, it doesn’t really do anything to further knowledge or provide an alternative method on MRI to identify metastatic disease other than by standard imaging criteria. 

Additionally, I disagree with the authors hypothesis that PMSA PET is less available than whole body MRI.  Whole body MRI is a very time intensive study and doesn’t really have a rationale as a separate modality to look for prostate cancer.  “The major limitation of PSMA PET is that it requires the administration of expensive radiotracers such as 68Ga-PSMA-11. As a result, it is not widely available outside of large, resource-rich medical centers equipped with adequate infrastructure, and this limits its utility. In contrast, while MRI is much more widely available, it misses a higher percentage of lesions compared to PSMA PET.”  This concept is grossly inaccurate and does not present an accurate or logical conclusion.  While the PSMA radiopharmaceutical the authors are using, (68Ga-PSMA-HBED-CC) is not FDA approved, there are several PSMA radiopharmaceuticals that are FDA approved have near universal distribution and availability in most parts of the United States, not just in large academic medical centers.  I doubt it is more feasible to have a whole-body MRI compared to an PSMA PET outside of a few unique cases.  Also, the authors claim “Ultimately, it is crucial to improve MRI interpretation because PSMA PET, though undoubtedly more powerful and superior to traditional methods, is currently limited by cost and availability to high-resource regions”.  Almost certainly, a whole-body MRI is much more expensive than a PSMA PET/CT (both out of pocket and with insurance) and has a much lower throughput.  And to emphasize this point again, the authors themselves claim whole body MRI is less accurate so why they are trying to justify a whole-body MRI alternative?

This study is well written, but it does not add to any knowledge or insight to the practitioner.  Specifically, while the authors note that 23% of cases involved human interpretive error, it does not specify what errors were made (e.g. wrong size?).  Thus, the practitioner is unable to use the information to improve their reads.  The authors report a “four-compartment signal model of whole-body diffusion-weighted imaging, which appears to have promising potential to evaluate prostate cancer bone metastases” but they did not attempt to use this model to reevaluate their data?  It would be more useful if perhaps the authors had re-interpreted their data with a unique system or even using an AI analysis to see if a computer could a unique commonality to metastatic nodes other than size.

Author Response

Dear MDPI Cancers Editors and Staff and Reviewers,

On behalf of my co-authors, I would like to thank you for taking the time to review our submission and provide comments. We hope that you will find it significantly improved, in no small part because of your thoughtful comments. 

The retrospective study, “Reasons for Discordance Between 68Ga-PSMA-PET and Magnetic Resonance Imaging in Men with Prostate Cancer” attempts to identify the cause of false negative MRI findings compared to PSMA PET in men with prostate cancer undergoing PSMA PET/MRI.  While this analysis could have utility to the reader and the field in the setting of local disease, visceral disease (e.g liver metastasis) or osseus lesions, the authors were only able to identify nodal metastatic lesions as the site of false negative finding on MRI.  The improved sensitivity and accuracy of PSMA PET compared to traditional imaging for nodal disease is a well-known and established aspect of molecular imaging (including other molecular PET agents for prostate cancer such as 11C-choline and 18F-fluciclovine) in that nodal metastatic disease is able to be identified on PSMA PET prior to pathological enlargement.  This is in part the excitement and adoption of PSMA PET/CT throughout the world.  Thus, while this analysis supports already available information in the literature, it doesn’t really do anything to further knowledge or provide an alternative method on MRI to identify metastatic disease other than by standard imaging criteria.

We agree, and we appreciate the reviewer’s insights. This manuscript is a sub-analysis of one of the studies showing the superiority of PET over MRI. Also, while it is well established that PET – PSMA or otherwise – is generally more sensitive that MRI for nodal disease, and while this study was not primarily powered to determine the superiority of PSMA PET over MRI for each kind of metastatic lesion, we could not find literature that had specifically, retrospectively examined a single prospective study to determine the kinds of lesions missed, nodal or otherwise, and the reasons underlying the superior performance of PET over MRI. We consider this the first step in determining where the weakness of MRI lie, in order to design methods (such as improved diffusion techniques, listed in the Discussion paragraph 6) to improve the value of MRI. To further elucidate the kinds of lesions missed, we have enumerated them in a new paragraph at the end of the Results.

Additionally, I disagree with the authors hypothesis that PMSA PET is less available than whole body MRI.  Whole body MRI is a very time intensive study and doesn’t really have a rationale as a separate modality to look for prostate cancer.  “The major limitation of PSMA PET is that it requires the administration of expensive radiotracers such as 68Ga-PSMA-11. As a result, it is not widely available outside of large, resource-rich medical centers equipped with adequate infrastructure, and this limits its utility. In contrast, while MRI is much more widely available, it misses a higher percentage of lesions compared to PSMA PET.”  This concept is grossly inaccurate and does not present an accurate or logical conclusion.  While the PSMA radiopharmaceutical the authors are using, (68Ga-PSMA-HBED-CC) is not FDA approved, there are several PSMA radiopharmaceuticals that are FDA approved have near universal distribution and availability in most parts of the United States, not just in large academic medical centers.  I doubt it is more feasible to have a whole-body MRI compared to an PSMA PET outside of a few unique cases. Also, the authors claim “Ultimately, it is crucial to improve MRI interpretation because PSMA PET, though undoubtedly more powerful and superior to traditional methods, is currently limited by cost and availability to high-resource regions”.  Almost certainly, a whole-body MRI is much more expensive than a PSMA PET/CT (both out of pocket and with insurance) and has a much lower throughput.  And to emphasize this point again, the authors themselves claim whole body MRI is less accurate so why they are trying to justify a whole-body MRI alternative?

This is a reasonable concern: we had not originally supported our contention that PET scanning is less widely available in the United States or worldwide. Therefore, we have added the following sentence to the introduction, with three citations to support it: “Especially given that there are an order of magnitude more MRI compared with PET units in the united states, with many states having only 2 or fewer centers providing this service, and with many countries in the “global south” lacking PET altogether, one cannot count on PSMA PET availability despite the FDA having approved more than one formulation.” Additionally, we agree that the cost of whole-body MRI may not necessarily be less than PSMA PET. We have therefore removed “cost” from that entry in the Discussion. We have also listed this in the (ultimate) limitations paragraph in the Discussion.

This study is well written, but it does not add to any knowledge or insight to the practitioner.  Specifically, while the authors note that 23% of cases involved human interpretive error, it does not specify what errors were made (e.g. wrong size?).  Thus, the practitioner is unable to use the information to improve their reads.  The authors report a “four-compartment signal model of whole-body diffusion-weighted imaging, which appears to have promising potential to evaluate prostate cancer bone metastases” but they did not attempt to use this model to reevaluate their data?  It would be more useful if perhaps the authors had re-interpreted their data with a unique system or even using an AI analysis to see if a computer could a unique commonality to metastatic nodes other than size.

We appreciate the reviewer’s complement, and agree that this does not provide much direct value over what is conventionally known: MRI is fairly sensitive for extranodal disease, and limited in terms of sensitivity for nodal disease when size and shape alone are used as discriminators. However, we may have been remiss in not stressing the specific interpretive errors made, and have added a paragraph to the end of the Findings delineating specifically those “atypical” and “undercalled” cases to guide interpretation and further research.

Reviewer 3 Report

Comments and Suggestions for Authors

The manuscript shed light on very important topic of PCA management, namely the role of PET PSMA in PCA staging. The data retrospectively analyzed relied on 38 patients with PCA.

The manuscript is well-written and easily readable. Specifically, it is noteworthy and valuable to report the characterization of "False negative" that the authors provided in the current paper. 

However, my concern with this classification is the Mischaracterization, defined as "The lesion was suspicious on mpMRI but was not initially reported." In my opinion, this statement was only misleading, due to the lack of details and inherent causes related to the mischaracterization process. First of all, the intervariability of the MRI interpretation. Second, the intravariability. If the authors may account for these two important causes, they could provide details. If the author might account for these two important causes, the "mischaracterization" should be sidelined. Additionally, just one patient exhibited mischaracterization. The authors relied on  sample size that prevent every generalization for the clinical practice.

Moreover, the population seems biased, due to the different stages of the disease they experienced. This aspect undermined the overall quality of the manuscript. Indeed, the authors compared naive patients suspected of PCA to PCA patients with BCR. Due to their different disease progression, the reliability of PET PSMA may drastically change. It should be stated in the limitation section. 

Author Response

Dear MDPI Cancers Editors and Staff and Reviewers,

On behalf of my co-authors, I would like to thank you for taking the time to review our submission and provide comments. We hope that you will find it significantly improved, in no small part because of your thoughtful comments.

The manuscript shed light on very important topic of PCA management, namely the role of PET PSMA in PCA staging. The data retrospectively analyzed relied on 38 patients with PCA.

The manuscript is well-written and easily readable. Specifically, it is noteworthy and valuable to report the characterization of "False negative" that the authors provided in the current paper.

Thank you for recognizing the potential importance of this topic and your kind assessment of our efforts.

However, my concern with this classification is the Mischaracterization, defined as "The lesion was suspicious on mpMRI but was not initially reported." In my opinion, this statement was only misleading, due to the lack of details and inherent causes related to the mischaracterization process. First of all, the intervariability of the MRI interpretation. Second, the intravariability. If the authors may account for these two important causes, they could provide details. If the author might account for these two important causes, the "mischaracterization" should be sidelined. Additionally, just one patient exhibited mischaracterization. The authors relied on  sample size that prevent every generalization for the clinical practice.

This is an important point. Although there was only one case of “mischaracterization,” or a lymph node which should have been described as suspicious based on size criteria but was missed, that only one such cases was identified out of 263 prospectively-enrolled subjects, it falls within normal radiologist accuracy. A 5th paragraph has been added to the Discussion to highlight this.

Moreover, the population seems biased, due to the different stages of the disease they experienced. This aspect undermined the overall quality of the manuscript. Indeed, the authors compared naive patients suspected of PCA to PCA patients with BCR. Due to their different disease progression, the reliability of PET PSMA may drastically change. It should be stated in the limitation section.

While the “limitations” (ultimate) paragraph in the discussion mentions that there were diverse indications, the risk of bias was not explicitly stated. This has now been appropriately included and we appreciate the opportunity to correct this omission.

Round 2

Reviewer 2 Report

Comments and Suggestions for Authors

Thank you for your responses.  No father edits are needed

Reviewer 3 Report

Comments and Suggestions for Authors

The authors answered properly my comments.